# Complementary Roles of Two DNA Protection Proteins from *Deinococcus geothermalis*

**DOI:** 10.3390/ijms24010469

**Published:** 2022-12-27

**Authors:** Min K. Bae, Eunjung Shin, Sung-Jae Lee

**Affiliations:** 1Department of Biology, Kyung Hee University, Seoul 02447, Republic of Korea; 2Department of Geography, Kyung Hee University, Seoul 02447, Republic of Korea

**Keywords:** *Deinococcus geothermalis*, chromosome stabiliser, DNA protection protein in starved cells (Dps), EMSA, iron sensing, oxidative stress response

## Abstract

The roles of two interrelated DNA protection protein in starved cells (Dps)—putative Dps Dgeo_0257 and Dgeo_0281—as orthologous proteins to DrDps1 for DNA binding, protection, and metal ion sensing were characterised in a *Deinococcus geothermalis* strain. Dgeo_0257 exhibited high DNA-binding affinity and formed a multimeric structure but lacked the conserved amino acid sequence for ferroxidase activity. In contrast, the Dgeo_0281 (DgDps1) protein was abundant in the early exponential phase, had a lower DNA-binding activity than Dgeo_0257, and was mainly observed in its monomeric or dimeric forms. Electrophoretic mobility shift assays demonstrated that both purified proteins bound nonspecifically to DNA, and their binding ability was affected by certain metal ions. For example, in the presence of ferrous and ferric ions, neither Dgeo_0257 nor Dgeo_0281 could readily bind to DNA. In contrast, both proteins exhibited more stable DNA binding in the presence of zinc and manganese ions. Mutants in which the *dps* gene was disrupted exhibited higher sensitivity to oxidative stress than the wild-type strain. Furthermore, the expression levels of each gene showed an opposite correlation under H_2_O_2_ treatment conditions. Collectively, these findings indicate that the putative Dps Dgeo_0257 and DgDps1 from *D. geothermalis* are involved in DNA binding and protection in complementary interplay ways compared to known Dps.

## 1. Introduction

Members of the genus *Deinococcus* are extremophiles that can survive a wide range of harsh conditions, including oxidative stress, desiccation, and extreme radioactive environments beyond the normal bacterial environmental adaptation zone [1,2,3]. *Deinococcus geothermalis* is 1 of more than 90 species of this highly radiation-resistant genus *Deinococcus*, which also includes the well-studied mesophile *D. radiodurans* (https://lpsn.dsmz.de/, accessed on 14 October 2022). *D. geothermalis* grows optimally at 48 °C and can survive under acute and chronic exposure to ionising radiation of up to 10 kGy at 60 Gy/h, 1 kJ/m^2^ of ultraviolet light, and years of desiccation. Furthermore, this bacterial species can also reduce Fe (III)-nitrilotriacetic acid, U (VI), and Cr (VI) and curtail Hg (II) in the presence of radiation and high temperatures and is, therefore, used in bioremediation [4,5,6,7,8,9].

Dps, a ferritin- and bacterioferritin-like protein subfamily, plays multifaceted roles in four major categories: (1) dodecameric assembly for iron storage and homeostasis, (2) DNA protection from oxidative damage, (3) bacterial genome packaging as nucleoid proteins, and (4) enzymatic activity [10,11,12]. These proteins can oxidise iron to prevent the formation of oxidative free radicals or form a protein–DNA complex to physically protect DNA [13]. Among the many resistance and protection mechanisms that enable *Deinococcus* to survive under high-stress conditions, we sought to characterise a DNA protection protein in starved cells (Dps) because of its crucial role in DNA damage protection, toxic metal ion storage and homeostasis, and reactive oxygen species (ROS) scavenging [10]. Dps has been described as a DNA-binding protein that regulates and protects the DNA of starved *Escherichia coli* [14,15]. *E. coli* Dps possesses nucleoid-clumping activity in *Staphylococcus aureus*, resulting in H_2_O_2_ resistance and enzymatic activity [16]. Furthermore, Dps also exhibits ferroxidase activity and can, therefore, catalyse the oxidation of ferrous iron (Fe^2+^) to its ferric state (Fe^3+^). H_2_O_2_ is reduced during this process, and Dps acts as an H_2_O_2_ detoxifier and DNA protector. Therefore, Dps is highly induced by direct H_2_O_2_ treatment; however, its expression can vary in a growth-phase- and species-dependent manner. The ability of microorganisms to resist metal toxicity is often related to their ability to express proteins capable of transforming metals into less soluble and less toxic chemical states [4]. *E. coli* Dps involved in copper homeostasis is a good example of this phenomenon [17]. A previous study reported that *D. geothermalis* can effectively decrease the cytotoxicity of Hg (II) and Hg (0) [4]. Similar results have been reported for Fe (III), U (VI), Tc (VII), and Cr (VI) under anaerobic and aerobic conditions [18]. Since *E. coli* Dps was first reported, various Dps have been reported in more than 10 bacterial genera (i.e., *Bacillus*, *Campylobacter*, *Lactobacillus*, *Helicobacter*, *Mycobacterium*, *Agrobacterium*, *Streptococcus*, *Pseudomonas*, *Vibrio*, and *Deinococcus*), as well as several archaeal Dps in *Sulfolobus solfataricus*, *Pyrococcus furiosus*, and *Halobacterium salinarum* [19,20,21,22,23].

The metal-ion-binding properties of bacterial Dps appear to correlate with the structure and function of iron storage proteins, such as ferritin (FtnA) and bacterioferritin (Bfr) [24,25]. Ferritin is a 24-mer protein with an inner diameter of 8 nm that can store approximately 24,000 Fe atoms in vivo [26,27,28]. However, the inner diameter of the 5 nm Dps provides only enough storage space for less than 500 iron atoms [19,21]. Thus, DNA-binding Dps can store iron in a bioavailable form and protect cells against oxidative stress. *E. coli* Dps has a shell-like structure and a spherical hollow cavity in the centre. The hollow cavity acts as an iron storage compartment, and iron sequestration is essential for iron detoxification and homeostasis. Oxidative protection is achieved by binding Fe^2+^ ions and preventing the Fenton-reaction-catalysed formation of toxic hydroxyl radicals or through DNA binding to prevent contact with oxidative radicals [29]. Although the basic mechanisms of iron entry and oxidation have been characterised, many essential questions regarding the mechanisms of iron core formation and release in response to cellular requirements have not yet been elucidated [30]. Zeth et al. recently studied the metal position of three ions (Co^2+^, Zn^2+^, and La^3+^) and the translocation pathway of *Listeria innocua* Dps [31]. Additionally, negatively charged residues (Asp and Glu) inside the cavity have been reported to cause a dynamic shift in the Dps conformation, resulting in metal ion storage by an electromagnetic force [24]. Thus, the Dps structure can be modified to specifically hold different metal ions for bioremediation purposes. Dps expression is predominantly induced both during starvation and in response to oxidative stress during the exponential phase [14,32,33]. In particular, Dps expression in *E. coli* is heavily dependent on the growth phase [15,34,35]. During exponential growth, Dps is upregulated by the OxyR regulator, which, in turn, is induced by H_2_O_2_ stress, and is activated by *σ*^70^-RNA polymerase. When exposed to H_2_O_2_ during the stationary period, Dps is induced by RpoS, encoded by *σ*^s^, and expressed as a dominant protein. In the absence of oxidative stress, Dps is downregulated by the nucleoid-associated proteins Fis and H-NS, which prevents the initiation of *dps* transcription by combining adjacent areas within the core *dps* promoter [15,36].

The VCO139 protein is strongly induced by exposure to H_2_O_2_ in *Vibrio cholera* [23]. This protein encodes a Dps homologous protein and is ROS-resistant. Furthermore, the expression of Dps is regulated by OxyR and RpoS during the exponential and stationary phases, respectively. Additionally, although this Dps can be expressed in the absence of H_2_O_2_, it is dramatically induced in the exponential growth phase but less induced in the stationary growth phase under H_2_O_2_. The involvement of Dps in the response to various environmental stressors has also been reported. For example, exposing the Δ*dps* strain to high iron levels during starvation decreases the number of viable Δ*dps* cells compared with that in the wild type. These data suggest that Dps is vital for the survival of bacteria in starved cells because of its reaction to ROS and its involvement in iron toxicity tolerance. Interestingly, *Campylobacter jejuni* Dps has an unusually high temperature tolerance, and its DNA-binding activity is activated by Fe^2+^. However, this DNA interaction can be inhibited by NaCl and Mg^2+^ [23,37]. In the case of the N_2_-fixing multicellular cyanobacterium *Nostoc punctiforme*, its genome contains five Dps genes involved in differential stress responses. For example, NpDps2 is the primary Dps involved in the light-induced oxidative stress response, and NpDps5 is involved in Fe^2+^ homeostasis. Furthermore, different NpDps are cooperatively activated for photosynthesis, N_2_ fixing, and stress response specificity [38]. In the hyperthermophilic crenarchaeon *Sulfolobus solfataricus*, Dps is directly induced by H_2_O_2_ treatment and Fe^2+^ depletion [20].

However, the expression of *dps* in *D. radiodurans* is repressed by OxyR, which contains one cysteine residue [39]. *D. radiodurans* possesses two Dps: 207 amino acid (aa)-long DrDps1 (DR2263) and 241-aa-long DrDps2 (DRB0092). DrDps1 and DrDps2 share only 16% amino acid sequence identity. However, both proteins have a typical dodecameric structure that forms a hollow spherical cavity [40]. The N-terminal region of DrDps1 contains a metal-binding site for iron and zinc, which is thought to be essential for regulating DNA–DrDps1 binding [41,42,43,44]. The first 30 amino acid residues of DrDps2 are thought to be signal peptides; therefore, this protein is likely located at the cell periphery and is protected from exogenous ROS in *D. radiodurans* [40,42]. Stress generation and H_2_O_2_ during the exponential cell growth phase can alter the oligomeric formation of DrDps1, which appears to be related to manganese homeostasis and ROS detoxification. However, H_2_O_2_ causes the accumulation of DrDps2 in its dodecameric cage form in the stationary phase, which could be considered a typical Dps response [40,44]. Furthermore, Dps can aggregate and remove metal ions that are known to be harmful to bacteria and humans. Therefore, the primary goal of our research is understanding the functional behaviour of Dps and Dps-like proteins in *D. geothermalis* for the industrial application of Dps.

Here, we focused on a putative Dps, Dgeo_0257, in *D. geothermalis* to compare the functional roles of the typical Dps response. Both Dgeo_0257 and Dgeo_0281 (DgDps1) were characterised using conformational analysis. Additionally, we evaluated the effect of different metal ions on the DNA-binding capacity of these proteins, as well as on cell growth/viability under H_2_O_2_ treatment. Furthermore, qRT-PCR analysis was conducted to characterise the expression of the two Dps genes at different growth phases during oxidative stress. Collectively, our findings indicate that the functions of the Dgeo_0257 protein are consistent with those of DNA-binding Dps in *D. geothermalis*, and its roles likely differ somewhat from those of DgDps1. However, the regulatory role of this protein remains speculative; therefore, additional studies are required to characterise its function in *D. geothermalis*.

## 2. Results

### 2.1. Characteristics of Amino Acid Sequences of a Putative Dps Dgeo_0257 and DgDps1

*D. geothermalis* possesses a 222-aa-long Dps1 orthologous gene (*dgeo*_0281) with 76.3% amino acid sequence identity to DrDps1 but lacks a gene equivalent to DrDps2 (Figure 1A). The putative Dps of the 200-aa-length Dgeo_0257 shared 65.5% amino acid sequence identity with the orthologous protein DR0582 as a hypothetical protein of *D. radiodurans* (Figure 1B). Both proteins exhibited different positions of a putative HTH domain in helices 4 and 5, according to Prabi prediction (http://www.prabi.fr/, accessed on 10 August 2022). Dgeo_0257 shared over 90% identity with a 194-aa DNA-binding protein from several *Deinococcus* genomes, including *D. apachensis*, *D. metallilatus*, *D. aerius*, and *D. phoenicis.* The *D. marmoris* genome contains two Dgeo_0257-like proteins that are 205 and 210 aa long, with 71% and 69% identities, respectively. However, despite the occurrence of these well-conserved specific proteins among the members of the genus *Deinococcus*, Dgeo_0257 has not yet been characterised and exhibited only 11.5% amino acid sequence identity with Dgeo_0281, a DgDps1 with ferroxidase activity (Figure 1C). At present, we cannot determine whether DgDps1 and Dgeo_0257 are convergent evolutionary products and have the same ancestor. Therefore, gene disruption and protein purification experiments should be conducted to determine the functional roles of Dgeo_0257, a DNA-protecting Dps lacking a conserved ferroxidase centre, compared with DgDps1.

### 2.2. Purification and Conformation of Dgeo_0257 and DgDps1 Proteins

Dgeo_0257 and DgDps1 proteins were overexpressed in *E. coli* by attaching C-terminal histidine tags, after which the proteins were purified by affinity chromatography using an Ni-NTA column. The profiles of Dgeo_0257 and DgDps1 proteins with a 6× His-tag were obtained using SDS-PAGE. Both proteins were induced using 1 mM isopropyl β-D-1-thiogalactopyranoside (IPTG); see Appendix A. The molecular weights of the Dgeo_0257 and DgDps1 monomers were approximately 23 kDa and 26 kDa, respectively. However, SDS-PAGE analysis revealed that Dgeo_0281 was over 30 kDa. Dgeo_0257 and DgDps1 proteins were purified at approximately 90% purity after Ni-NTA affinity chromatography per litre of culture in the first stage of the one-step purification process at 5 mg and 20 mg, respectively (Appendix A).

Proteins of various sizes were collected and confirmed using gel filtration. Dgeo_0257, a 23 kDa monomer with His-tag, was expected to form a 12-mer or 24-mer polymer or a sphere of 240 kDa or 480 kDa, similar to DrDps1 and DrDps2 of *D. radiodurans* [40]. Gel filtration analysis results indicated that the intact protein elution times of Dgeo_0257 were 20.25 and 23.34 min, which may correspond to two multimeric conformations, despite the Superdex 75 column being limited to efficiently correlate until a 70 kDa separation range; a 440 kDa size marker revealed a 21.39 min elution peak. However, the elution peaks of DgDps1 were exhibited at 25.04 and 28.32 min, which suitably corresponded to the dimer and monomer forms (Appendix A). The conformation did not affect the elution of the Dgeo_0281 protein in the presence of 1 mM iron. However, the elution fraction of Dgeo_0257 fully or partially moved to the posterior region when 1 mM Fe^2+^ or Fe^3+^ was present. Therefore, Dgeo_0257 protein conformation was altered by the iron redox state in multimeric forms but did not exhibit dimeric or monomeric forms. The conformational changes in Dgeo_0257 need to be precisely defined by gel filtration using a Superdex 200 column and physical gradient analysis using sucrose density.

### 2.3. DNA-Binding and Protection Properties of Dgeo_0257 and DgDps1 Proteins

In general, bacterial Dps have nonspecific DNA-binding activity [14,15]. To confirm this phenomenon, we performed electrophoretic mobility shift assay (EMSA) detection using 6% native polyacrylamide gel electrophoresis with a ca. 250-nt-length promoter and an open reading frame (ORF) region. Both Dgeo_0257 and DgDps1 showed distinct DNA-binding affinities and DNA–protein complex positions with two different DNA fragments of the promoter and ORF regions of randomly selected Dgeo_2112 (Figure 2). The DNA–protein complex was in a super-shifted position in Dgeo_0257 (Figure 2A). However, the DNA–protein complex in DgDps1 was at a lower position (Figure 2B). The unique DNA-binding affinity of the two putative Dps was identified through half-maximal DNA–protein complex formation. The DNA-binding activity of Dgeo_0257 was over five-fold higher than that of DgDps1: the Dgeo_0257 protein required a concentration of 10 μM for complete shift-up migration, whereas DgDps1 required a protein concentration of >60 μM for full DNA binding. Interestingly, the DNA-binding behaviour of Dgeo_0257 and DgDps1 differed between the promoter and ORF regions. Treatment with Fe^2+^ at 0.2 and 1 mM resulted in complete release of the promoter region DNA from the Dgo_0257 and DgDps1 protein–DNA complexes, respectively. However, Dgeo_0257 exhibited free DNA and a lower position band, whereas DgDps1 showed better binding to the ORF region DNA at a lower position (Figure 2). To compare the effects of Fe^2+^ and Fe^3+^ on DNA–protein complex formation, the constant concentrations of Dgeo_0257 and DgDps1 protein were treated with 10 µM and 30 µM, respectively, and increasing amounts of iron ions were applied (Figure 2C,D). Dgeo_0257–DNA complexes dissociated at 0.2 mM and 1 mM, and DgDps1–DNA complexes dissociated at 1 mM and 3 mM of Fe^2+^ and Fe^3+^, respectively. Interestingly, although 5 mM Fe^3+^ completely released free DNA, increasing the concentrations of Fe^2+^ affected the stability of free DNA. In DNA protection of both Dgeo_0257 and DgDps, after free DNA was released from the DNA–protein complexes, the free DNA gradually disappeared at 1.0 mM and 3 mM of Fe^2+^ on the ORF regions (lanes 8–10 in Figure 2C).

To examine the DNA-protective activity of Dps, we found that free DNA was no longer detectable in the presence of 0.5 mM of Fe^2+^ (Figure 3). This was because unprotected naked DNA was damaged and degraded owing to oxidative stress through Fenton reactions, which led to the production of hydroxyl radicals from ferrous ions or ferrous ions plus H_2_O_2_ mixtures (lanes 2, 6, and 10 in Figure 3). However, the DNAs were well protected from oxidative stress with the addition and increased levels of Dgeo_0257 and DgDps1 proteins (lanes 4, 5, 8, and 9 in Figure 3). Therefore, Dgeo_0257 and DgDps of *D. geothermalis* are DNA-binding proteins with nonspecific DNA sequence targets that possess distinct patterns with Fe^2+^ in both the promoter and ORF regions. More importantly, these proteins protect DNA from oxidative stress reactions induced by H_2_O_2_ treatment.

### 2.4. Effect of Metal Ion Sensing on the DNA-Binding Activity of a Putative Dps Dgeo_0257 and DgDps1

Next, we sought to characterise the effects of various metal ions on Dps–DNA-binding activity (Appendix A). With increased Fe^2+^ concentrations, the DNA–protein complex did not maintain a complex form, and free DNAs were released from both Dgeo_0257 and DgDps1–DNA complexes at Fe^3+^ levels exceeding 1 mM and 3 mM, respectively (Figure 2C,D). Similar to Fe^2+^, Fe^3+^ induced similar DNA-releasing activity, albeit at more than 2–3-fold higher concentrations (Figure 2D). Free DNA was drastically degraded by high concentrations of Fe^2+^ (lanes 9 and 10 in Figure 2C). Interestingly, as the concentrations of manganese and zinc ions increased to 5 mM, the DgDps1 and Dgeo_0257 proteins much better bound to DNA, which further stabilised DNA binding (Figure 4). Interestingly, in the presence of more than 2 mM of Mn, a novel DgDps1 protein–DNA complex was observed (Figure 4). The reaction with manganese was consistent with previous reports from different research groups [41,42]. However, to the best of our knowledge, this study is the first to characterise the effect of zinc on DNA-binding stabilisation. Lead, cobalt, and copper did not affect DNA-binding activity (Appendix A). However, DNA-binding activity varied with chromate and caesium, depending on the metal ion concentration. The DgDps1 protein showed fluctuating patterns, depending on the specific concentrations of caesium (Appendix A). In contrast, caesium did not affect the DNA-binding activity of Dgeo_0257 but chromate did in a concentration-dependent manner. DgDps1 was released from the DNA–Dps complex by chromate at over 0.1 mM. Therefore, although the amino acid sequence, conformation, and DNA-binding affinity equivalence between the putative Dps Dgeo_0257 and DgDps1 were quite different, both proteins exhibited similar DNA-binding behaviours in the presence of metal ions, except for caesium and chromate.

### 2.5. Physiological Properties of Dps-Gene-Disrupted Mutants

The disrupted mutant strains of a putative Dps *dgeo*_0257 and *dgeo*_0281 (Δ*dgeo*_0257 and Δ*dgeo*_0281) were constructed by homologous recombination and kanamycin-resistant selection in our previous study [45]. The Δ*dgeo*_0257 mutant strain adapted well to 100 mM but did not grow at 120 mM, although the DgDps1-deficient strain (Δ*dgeo*_0281) exhibited higher sensitivity (Figure 5A). In the absence of oxidative stress, the two mutant strains showed an almost identical growth pattern (Figure 5B). This means that both DgDps1 and putative Dps *dgeo*_0257 genes are not essential. Following direct H_2_O_2_ treatment experiments, the WT and two Dps-gene-disrupted mutants were treated with 100 mM H_2_O_2_ before reaching OD_600_ = 0.5 in direct measurements, and all strains did not grow (Figure 5B). In the mid-exponential growth stage (OD_600_ = 1.47 in direct measurements), each strain showed different levels of growth inhibition upon treatment with 100 mM H_2_O_2_ (Figure 5C). The viability test was performed to confirm the oxidative stress response among the wild-type and Δ*dgeo*_0257 and Δ*dgeo*_0281 mutant strains after treatment with H_2_O_2_ at different concentrations (80, 100, and 120 mM). The WT strain grew well even in 120 mM H_2_O_2_, whereas the Δ*dgeo*_0281 mutant strain showed a slight decrease in viability at 80 mM, and cells did not grow above 100 mM H_2_O_2_. As expected, the complemented strains for *dps*-deficient mutants overcame growth inhibitions, and the viability assay under H_2_O_2_ treatment exhibited a rescue in the Δ*dgeo*_0281 complementary strain (Appendix A). Therefore, our findings confirmed that both Dps-related proteins participate in the antioxidation reaction.

### 2.6. Expression Levels of Dgeo_0257 and DgDps1 Genes through Antiparallel Influence in a Growth-Phase-Dependent Manner

Experiments were performed to compare the expression levels of both *dps* genes *dgeo*_0281 (DgDps1) and *dgeo*_0257 in the wild-type strain and Δ*dgeo*_0257 and Δ*dgeo*_0281 mutants, respectively. The expression level of *dgeo*_0257 at OD_600_ 2.0 in the wild-type strain was used as the control. In the early stages of growth of wild-type *D. geothermalis*, DgDps1 was highly upregulated (over a 150-fold increase); however, DgDps1 levels decreased sharply as the growth phase progressed. Furthermore, H_2_O_2_ did not dramatically affect the expression of DgDps1 during the growth phase. However, the expression of DgDps1 was enhanced at an OD_600_ of 4.0 in the presence of H_2_O_2_ (Figure 6A). Moreover, the expression level of *dgeo*_0257 gradually increased (2.46-fold increase) when the cells reached the stationary phase. Dgeo_0257 expression reached its maximum (11.63-fold increase) after treatment with 50 mM H_2_O_2_ at an OD_600_ of 4.0 (Figure 6B). Therefore, we concluded that DgDps1 plays a significant role in the early exponential growth phase without oxidative stress. Furthermore, both DgDps1 and Dgeo_0257 were highly expressed under oxidative stress, particularly Dgeo_0257 in the late exponential growth phase, and both were expressed with an antiparallel influence on a growth-phase-dependent manner in the absence of oxidative stress in *D. geothermalis*.

The Δ*dgeo*_0257 mutant showed a half-maximum expression level of DgDps1 (60.69-fold increase) at the beginning of the growth phase. However, these levels gradually decrease in the absence of oxidative stress. Moreover, the expression level of DgDps1 increased two-fold at both OD_600_ 4.0 and 8.0 under oxidative stress conditions (Figure 6C). Thus, DgDps1 takes over the role of Dgeo_0257 during the growth phase because of the absence of *dgeo*_0257 under oxidative stress, which may be involved in DNA protection. In contrast, the Δ*dgeo*_0281 mutant exhibited higher Dgeo_0257 expression (more than 10-fold) than the wild type at OD_600_ 2.0. Thus, we inferred that Dgeo_0257 expression increased to compensate for the absence of *dgeo*_0281 until an OD_600_ of 4.0 was reached (Figure 6D). The antiparallel expression of both Dps genes in each gene-deficient mutant reduced in complementary strains under oxidative stress (Appendix A). Therefore, the expression of DgDps1 and Dgeo_0257 appears to depend on that of other DNA-protecting and ROS-scavenging proteins to ensure cell survival under oxidative stress conditions. In this way, one gene compensates for the changing expression patterns of the other genes throughout the different growth phases in *D. geothermalis*.

## 3. Discussion

Dps is a known DNA-binding protein without sequence specificity [15]. However, Chip-seq analysis demonstrated that *E. coli* Dps combines with genomic DNA in a non-random manner [46]. DgDps1 (Dgeo_0281) and Dgeo_0257 also exhibit different DNA-binding specificities in the selected promoter and ORF regions (Figure 2 and Figure 4). Owing to the self-assembly and DNA-binding properties of Dps in solution, these proteins become extensively agglomerated through co-crystallisation and general defence mechanisms [47].

Dps forms a dimer or dodecamer structure and binds to DNA [48]. The oligomeric states of the DrDps1 protein of *D. radiodurans* change depending on the growth phase and presence of DNA. This suggests that the oligomeric forms and functional roles of DrDps1 may adapt to environmental changes [49]. In contrast, the DrDps2 oligomeric form maintains a dodecameric structure without affecting the surrounding environment. DrDps2 is thought to be more selective towards iron, depending on the cellular environment, and is involved in intracellular metal storage [40,42]. In this study, the protein form of Dgeo_0281 (DgDps1), the homologous protein of DrDps1, was not identified in its dodecameric form. Instead, this protein exhibited a dimeric structure according to gel filtration and EMSA tests (Appendix A and Figure 2B). The DNA-binding affinity was Kd > 40 μM, which was half the maximum of the nonspecific DNA sequence. Interestingly, the putative Dps Dgeo_0257 showed higher DNA affinity than DgDps1, with a Kd value of <4 μM. EMSA showed that the Dgeo_0257–DNA complex was located near the upper pocket of the gel, unlike DgDps1 found below (Figure 2A). Furthermore, the two proteins Dgeo_0257 and DgDps1 exhibited similar electrophoretic shifts with DNA in the presence of all metal ions evaluated herein, except for caesium and chromate, as demonstrated by EMSA (Appendix A). Thus, additional studies are required to analyse the effects of metal ions on the protein structure and DNA binding.

Therefore, we now focused on determining whether Dgeo_0257 and DgDps1 can react with these heavy and harmful metal ions. When manganese and zinc ions were present, both DgDps1 and Dgeo_0257 exhibited more stabilising DNA binding for DNA protection. This correlates well with Daly’s hypothesis that manganese acts as an antioxidant [5,6]. The behaviour of zinc ions as inducers of gene expression of the RDR regulon was exhibited by DdrO cleavage by an activated metalloprotease IrrE [50]. However, in the presence of zinc ions, both Dps had enhanced DNA-binding capacity (Figure 4). Owing to their ability to act as protein cages for iron and various other metals, Dps-like proteins have recently garnered increasing attention in the field of nanotechnology [8,13].

We previously conducted RNA-seq to define the functional roles of the putative Dps Dgeo_0257 (NCBI GEO accession number GSE151903) [45]. Interestingly, 36 proteins/enzymes and IS*Dge5* transposases were upregulated, and 19 genes were downregulated. Based on these results, we concluded that the putative Dps Dgeo_0257 is involved in specific gene regulation, as well as DNA protection and stabilisation under oxidative stress conditions.

This study was conducted under the assumption that DgDps1 shares similar characteristics with existing Dps, and Dgeo_0257 has similar Dps characteristics and lacks a ferroxidase active centre. Interestingly, both Dgeo_0257 and DgDps1 proteins showed similar metal ion sensing. However, the putative Dps Dgeo_0257 exhibited a six-fold higher DNA-binding affinity and higher metal ion sensitivity than DgDps1 (Figure 2). Our findings suggest that Dgeo_0257 and DgDps1 proteins are responsible for intracellular DNA protection and detoxification of harmful iron in a growth-phase-dependent manner. The expression of DgDps1 was dominant during the early exponential growth phase and was not induced during the stationary growth phase in the absence of H_2_O_2_. In the presence of H_2_O_2_, DgDps1 levels in the Dgeo_0257-disrupted mutant were gradually induced by cell growth. However, Dgeo_0257 was sensitive to H_2_O_2_-induced oxidative stress and was induced in the early exponential growth phase when the dominantly expressed DgDps1 was absent (Figure 6). Therefore, we propose that Dgeo_0257 is a putative Dps DNA-protecting protein in *D. geothermalis* and plays antiparallel protection functions to that of DgDps1. Additional studies are required to characterise the Dps function and conformational structure of Dgeo_0257, as well as its role in network regulation through redox-sensing regulators and signalling molecules in *D. geothermalis*.

## 4. Materials and Methods

### 4.1. Bacterial Strains and Culture Condition

The *D. geothermalis* KACC 12208 strain derived from DSM 11300 ^T^ was obtained from the Korean Agricultural Culture Collection (KACC, http://www.genebank.go.kr; accessed on 21 December 2008). The Dps gene-disrupted mutants Δ*dgeo*_0257 and Δ*dgeo*_0281 were constructed in a previous study via homologous recombination of the contiguous regions of the kanamycin resistance gene in the pKatAph3 vector to determine cell viability and growth patterns, as well as to conduct RNA-seq analysis (Appendix A) [45,51]. Complemented strains, including a shuttle vector for gene expression, were constructed in this study. The genes encoding Dgeo_0257 were recombined into the shuttle vector pRadgro and transformed into each gene-deficient strain [52].

*E. coli* DH5α was routinely used for cloning and protein purification procedures. *E. coli* strains containing cloned recombinant plasmids were cultivated in LB broth medium containing 100 µg/mL of ampicillin at 37 °C. For protein expression at the late exponential phase (ca. OD_600_ 0.8, directly measured), the *lac* promoter of *E. coli* cells was induced with 1 mM IPTG (final concentration) and continuously cultured for 4 h. After this 4 h induction, the culture was centrifuged at 5000 rpm for 20 min and the supernatants were discarded. The pellets were stored at −70 °C until required for downstream experiments. *D. geothermalis* was cultured at 48 °C in TGY medium containing 1% tryptone, 0.5% yeast extract, and 0.1% glucose at 48 °C. *E. coli* cells were cultured at 37 °C in Luria-Bertani (LB) medium (MB Cell, Kisan Bio, Seoul, Republic of Korea) containing 1% tryptone, 0.5% yeast extract, and 1% NaCl.

### 4.2. Construction of an Expression Vector for Dgeo_Dps Genes

The *dgeo*_0257 and *dgeo*_0281 genes were constructed and amplified using the PCR products encompassing both genes with different restriction enzyme sequences, double-digested with *Nco*I and *Bgl*II for *dgeo*_0257 and with *Nco*I and *Bam*HI for *dgeo*_0281, and ligated into the pCS19 vector to obtain carboxyl-terminal His-tag versions of the target proteins (Appendix A). Next, the recombinant plasmids were transformed into *E. coli* DH5α competent cells using the CaCl_2_ chemical transformation method. The transformants were then selected on a 100 µg/mL ampicillin-containing plate. Correct construction was detected using colony PCR with appropriate gene-specific primer pairs and finally determined by DNA sequence analysis using an ABI 3730xl capillary electrophoresis sequencing system (Macrogen, Seoul, Republic of Korea).

### 4.3. Purification of the Putative Dps Protein (Dgeo_0257) and DgDps1 (Dgeo_0281)

Cell pellets stored in a −70 °C freezer were resuspended with 5 mL lysis buffer (300 mM NaCl, 50 mM NaH_2_ PO_4_, pH 8.0, and 10 mM imidazole) per pellet obtained from 250 mL. Cell membranes were broken using a French pressure cell press (with 7000 psi; Vision Biotech, Gimpo, Republic of Korea) or an ultra-sonicator (30% maximum potency 20 s on and 40 s off for three cycles, followed by 21% maximum potency; CV334 ultrasonic processor; Sonics & Materials Inc., Newtown, CT, USA). Following cell lysis, the samples were centrifuged at 5000 rpm for 20 min, and the supernatants were collected for further testing. Over-expressed C-terminal His-tagged target proteins were purified using nickel-nitrilotriacetic acid (Ni-NTA; Qiagen, Hilden, Germany) resin column chromatography. After equilibrating the Ni-NTA resin column with lysis buffer, the sample solution was loaded and the pass-through was collected. Next, 10–20 mL of washing buffer (300 mM NaCl, 50 mM NaH_2_ PO_4_, pH 8.0, 20 mM imidazole) was added. Finally, 10 mL of elution buffer (300 mM NaCl, 50 mM NaH_2_ PO_4_, pH 8.0, 250 mM imidazole) was poured into the resin column to recover the target proteins. Protein production was then confirmed using Bradford assay, after which the target protein profiles were characterised via 12% SDS–polyacrylamide gene electrophoresis (PAGE). Electrophoresis was performed in 1× tank buffer (25 mM Tris, pH 8.3, 192 mM glycine), after which the gel was placed in staining buffer with 0.1% Coomassie Blue R250 in 10% acetic acid, 50% methanol, and 40% distilled water (DW) for at least 3 h (usually left overnight). Destaining (10% acetic acid, 50% methanol, and 40% DW) was performed to visually confirm the protein bands of interest. Protein dialysis was performed using a cellulose tubular membrane (CelluSep H1; Membrane Filtration Products Inc., Seguin, TX, USA) with 1x dialysis solution (25 mM Tris, pH 7.5, 150 mM NaCl). The membrane was placed in 1 L of dialysis solution and was rotated to ensure that it was fully submerged in the buffer for 1 h. The process was repeated three times, after which the protein concentration was determined following the VivaSpin-6 manufacturer’s guidelines (7000 rpm for 15–20 min; Sartorius, Göttingen, Germany).

### 4.4. Determination of Protein Conformation by Gel Filtration

Size-exclusion chromatography (SEC) was performed using the BioLogic HR Chromatography system (Bio-RAD, Hercules, CA, USA) with a HiLoad 16/600 Superdex 75 prep grade column (GE Healthcare Life Sciences, Chicago, IL, USA). The SEC column was filled with a running buffer solution (25 mM Tris-HCl, pH 7.5, 150 mM NaCl) to separate the molecules adhered throughout the resin loaded in the column. The buffer was then passed through the column at a 2.0 mL/min flow rate for 2 h. Protein size markers, including ferritin (440 kDa), conalbumin (75 kDa), ovalbumin (43 kDa), and anhydrase from erythrocytes (29 kDa), were loaded first, followed by the purified proteins at a 0.5 mL/min flow rate for 1 min (Sigma-Aldrich, St. Louis, MO, USA). The running buffer was then drained at a 2.0 mL/min flow rate for 1 h and dispersed in the fraction collector. Aliquots of critical parts that exceeded pre-established threshold values were collected, concentrated with a VivaSpin-6 protein concentrator column, and used in downstream experiments.

### 4.5. Electrophoretic Mobility Shift Assay (EMSA)

Both the promoter and open reading frame (ORF) regions of Dgeo_2112 as a hypothetic protein were amplified from *D. geothermalis* genomic DNA by PCR using forward and reverse primer sets. For the promoter region, the forward and reverse primer sequences were “ATCCTTCGCTCTCGTTGGAA” and “TTTTGGTGGAGACAGGCAGA”, respectively. For the ORF region, the forward and reverse primer sequences were “TGTTAAGCCGTTCAACCCGA” and “TTCAAACTGGGTCGGCAACA”, respectively. For EMSA, DNA probes approximately 250 nucleotides in length were purified using the AccuPrep^®^ PCR purification kit (Bioneer, Daejeon, Republic of Korea). Specific amounts of DNA and Dgeo_0257 protein were mixed in 5× EMSA binding buffer (100 mM Hepes (pH 7.2), 160 mM KCl, 0.5 mM EDTA (pH 8.0), 50% glycerol, 350 ng/μL of BSA, 40 ng/μL of poly(dl-dC), 12.5 mM DTT) in 10 μL volumes. The prepared metal ions included Fe (II) from ammonium iron (II) sulphate hexahydrate, (NH_4_)_2_ Fe(SO_4_)_2_.6H_2_ O (Sigma-Aldrich, St. Louis, MO, USA), and Fe (III) from ammonium iron (III) citrate, C_6_H_8_O_7_.Fe.NH_3_ (Sigma-Aldrich, St. Louis, MO, USA), in addition to other metal ions, including Zn (II), Mn (II), Cr (VI), Co (II), Cu (II), Cs, and Pb (II). The DNA–protein-binding reactions were performed at 37 °C for 30 min and loaded onto 6% native polyacrylamide gels buffered in 0.5× Tris-borate-EDTA (TBE). After conducting PAGE at 60 V for 120 min, the DNA in the gels was stained with SYBR^TM^ Gold nucleic acid gel staining solution (Life Technologies Co., Carlsbad, CA, USA). The band positions and intensities were then detected using a Chemi-Doc imaging system (Bio-RAD, Hercules, CA, USA).

### 4.6. qRT-PCR Analysis of Wild-Type and Δdgeo_0257 and Δdgeo_0281 Mutant Strains at Different Growth Phases

After overnight incubation, both wild-type and *dps*-disrupted mutant strains were pre-cultured. Afterwards, 5 mL of TGY broth was adjusted with differential inoculation sizes based on the delayed growth patterns of the mutant strains (approximately 1 h). After 3 h, optical density measurements were performed at OD_600_, adjusted to an OD_600_ value of 0.06, and the main culture was started in 50 mL of TGY. Upon reaching an OD_600_ value of 2.0, 4.0, and 8.0, the samples were centrifuged in 5 mL, 2.5 mL, and 1.25 mL, respectively, to match the cell numbers, after which the supernatants were discarded, and the cell pellets resuspended in 5 mL of 0.9% NaCl and then treated with 50 mM H_2_O_2_ for 30 min. After the H_2_O_2_ challenge, the samples were centrifuged at 10,000 rpm for 5 min and washed once more with 0.9% NaCl. The supernatants were discarded, and the pellets were stored at −20 °C. Samples without H_2_O_2_ treatment were used as a control.

The cell wall was broken using phenol to determine the gene expression level. DNA digestion was performed using DNase I, and RNA was extracted using an RNA prep kit for RNA cleanup (RNeasy mini purification kit; Qiagen, Germany). After measuring the extracted RNA concentration, the concentration was normalised to 1000 ng in 8 μL volume for all samples. cDNA synthesis was performed with a dNTP mixture and 6-mer random primers using the following protocol: 60 °C for 5 min, 4 °C for 3 min, 30 °C for 10 min, 42 °C for 60 min, and finally 95 °C for 5 min (PrimeScript^TM^ 1st strand cDNA Synthesis Kit; TaKaRa, Japan). In the step at 4 °C for 3 min, 4 μL of 5× buffer, 4.5 μL of RNase free water, 1 μL of RTase, and 0.5 μL of RNase inhibitor were added. Each *dgeo*_0257- and *dgeo*_0281-specific primer pair was then added to the obtained cDNA. qRT-PCR analysis was performed using TB Green^®^ Premix Ex Taq^TM^ (TaKaRa, Osaka, Japan) on a Bio-RAD RT-PCR model CFX96 ^TM^ Optics Module (Bio-RAD, Hercules, CA, USA). The expression of the target genes was normalised to that of GAPDH, a constitutively expressed gene throughout all growth phases. The related expression levels of both *dps* genes were calculated, as described in a previous study [41]. Differences between samples were determined via Student’s *t*-test using Prism^TM^ ver. 8.0 software. Differences were deemed statistically significant at *p* < 0.05 (*), *p* < 0.01 (**), *p* < 0.001 (***), and *p* < 0.0001 (****).

## 5. Conclusions

Dps has a general DNA-protective role under oxidative stress conditions in various prokaryotes. Dps forms multifaceted structural conformations for Fe^3+^ sequestration, resulting in detoxification against the cellular Fenton reaction and homeostasis of Fe^3+^ and others (Figure 7). Here, we characterised DgDps1 (Dgeo_0281) and a putative Dps candidate protein from the radiation-resistant bacterium *D. geothermalis*. The putative Dps Dgeo_0257 has 10-fold higher DNA-binding activity than DgDps1, and the presence of Fe^3+^ and Fe^2+^ results in DNA release, with similar behaviour to DgDps1. In the presence of manganese and zinc ions, both DgDps1 and Dgeo_0257 exhibit more stable DNA binding for DNA protection. Both DgDps1 and Dgeo_0257 are antiparallelly expressed in a growth-phase-dependent manner in the absence of H_2_O_2_ conditions. Notably, DgDps1 is dominant in the early growth phase, and Dgeo_0257 is gradually induced from the early phase to the stationary phase. However, when 50 mM H_2_O_2_ is present, their expression ratios are enhanced in each other’s gene-deficient mutant: DgDps1 is drastically induced at the stationary phase in the Dgeo_0257-deficient mutant, and Dgeo_0257 is strongly induced at the exponential growth phase in the DgDps1-deficient mutant. Therefore, the transcriptional regulation of both Dps is growth-phase-dependent and complementary to each other.

## Figures and Tables

**Figure 1 ijms-24-00469-f001:**
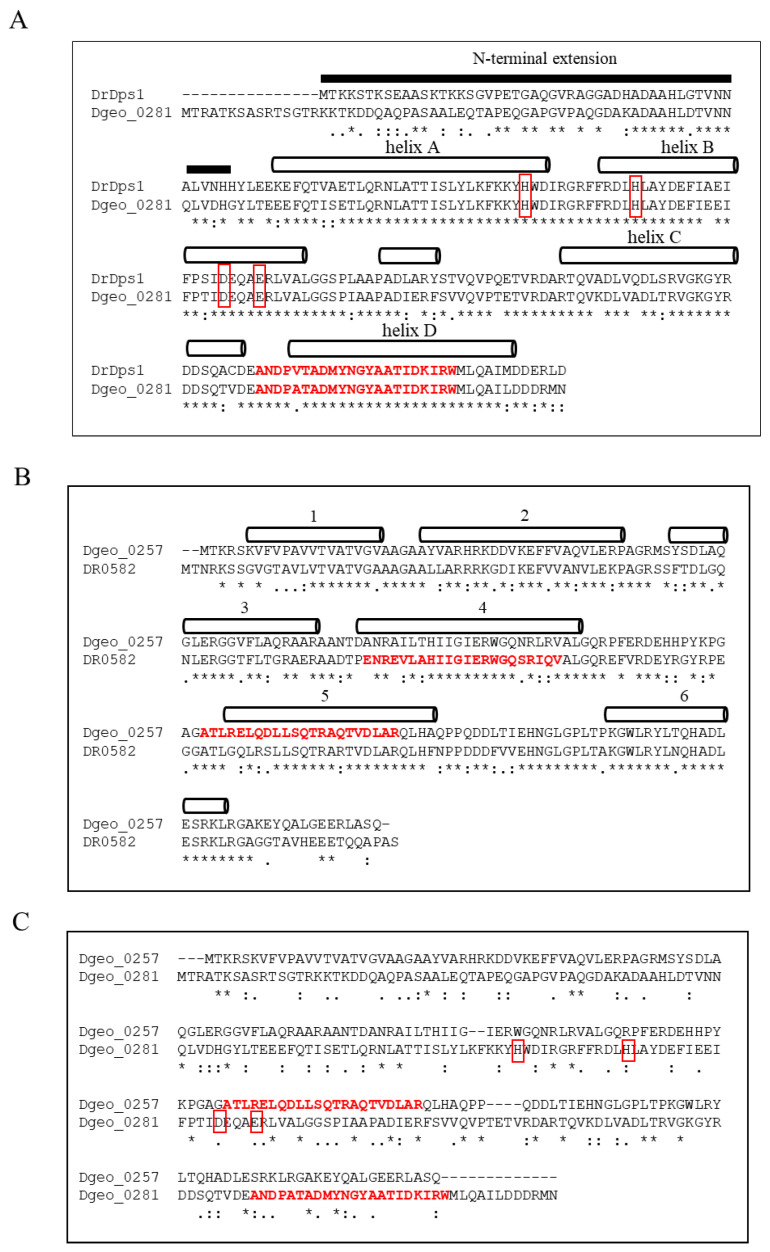
Comparison of amino acid sequences between DrDps1 and DgDps1 (Dgeo_0281) (**A**), a putative Dps Dgeo_0257 of *D. geothermalis* and a hypothetic protein DR0582 of *D. radiodurans* with 65.5% identity (**B**) and between DgDps1 and Dgeo_0257 with 11.5% identity (**C**). The secondary structure of the helix (**A**,**B**) and the ferroxidase active centre of DgDps1 with red boxes are marked (**A**,**C**). The conserved ferroxidase active centre is missing in Dgeo_0257 (**C**). Red words indicate the DNA-binding domains (HTH) predicted by Prabi (https://npsa-prabi.ibcp.fr/ accessed on 10 June 2022). The secondary structure of Dgeo_0257 was predicted by Jpred 4 (http://www.compbio. dundee.ac.uk/jpred4/ accessed on 1 June 2022).

**Figure 2 ijms-24-00469-f002:**
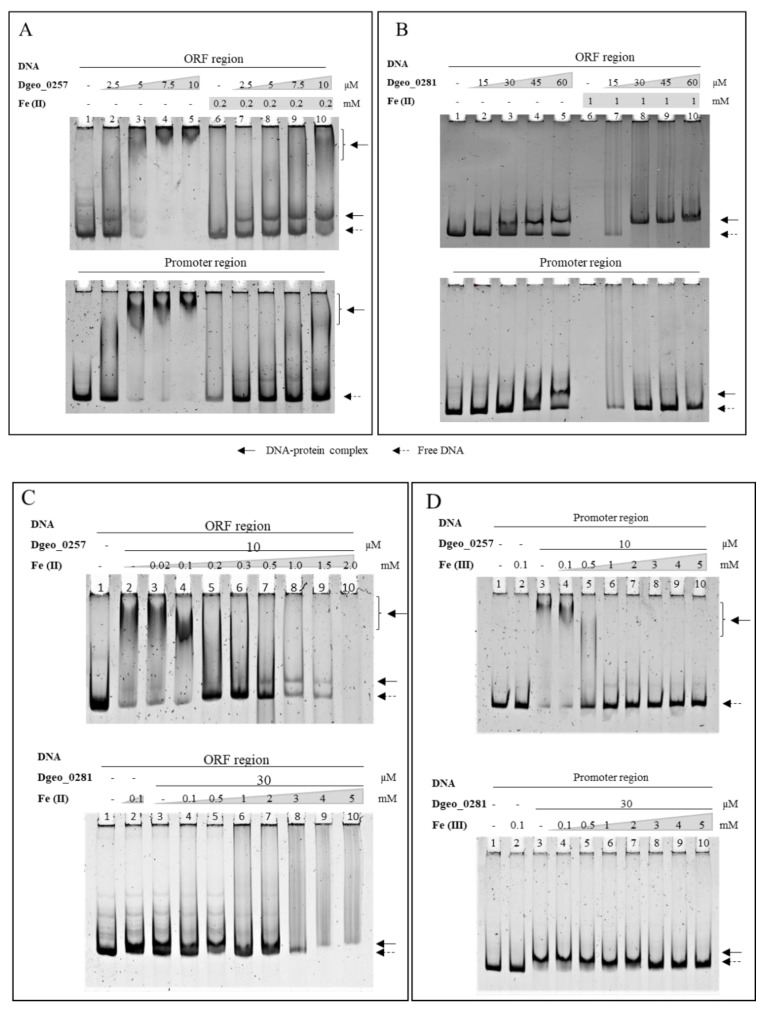
EMSA of Dgeo_0257 (**A**) and DgDps1 (**B**) on both promoter and ORF regions with increasing protein concentration. Dgeo_0257 and DgDps1 were treated with 0.2 mM and 1.0 mM ferrous ion concentration at lanes 6–10, respectively. Lane 1, free DNA; lanes 2–5, increased amounts of purified Dgeo_0257 and DgDps1 proteins (2.5–10 µM and 15–60 µM, respectively). Comparison of differential concentrations of ferrous ions (**C**) and ferric ions (**D**) on ORF and promoter regions, respectively.

**Figure 3 ijms-24-00469-f003:**
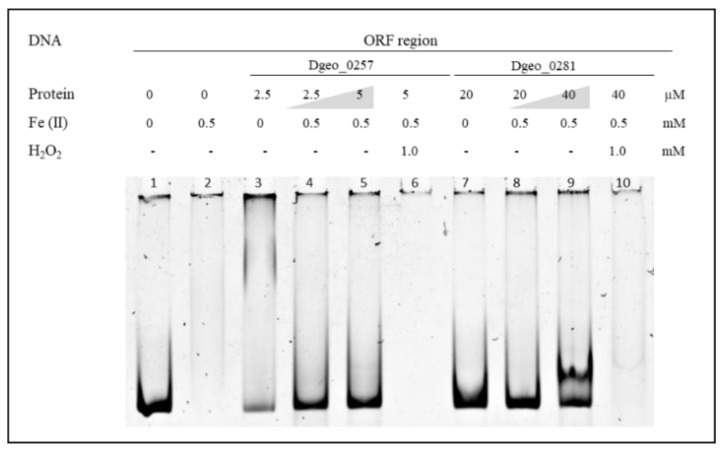
DNA–Dps complex formation and DNA protection analysis from iron treatment and Fenton reactions. Lane 1, free DNA; lane 2, only 0.5 mM Fe^2+^; lanes 3 and 7, both proteins added with different amounts; lanes 4, 5, 8, and 9, increased Dgeo_0257 and DgDps1 proteins with ferrous ions at 0.5 mM; lanes 6 and 10, Fenton reaction samples with 1.0 mM H_2_O_2_.

**Figure 4 ijms-24-00469-f004:**
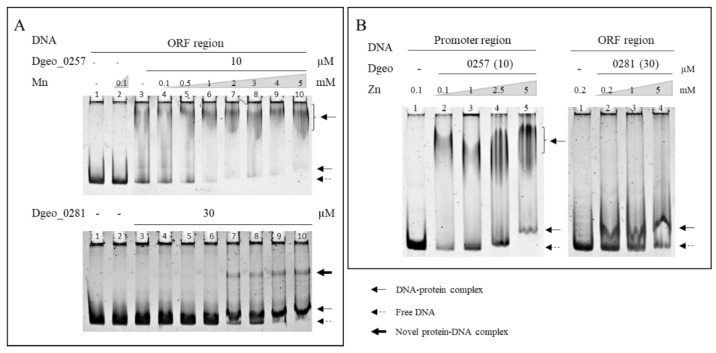
EMSA of Dgeo_0257 and DgDps1 maintaining a protein concentration of 10 µM and 30 µM, respectively, while increasing the metal ion concentration of manganese (**A**) and zinc (**B**).

**Figure 5 ijms-24-00469-f005:**
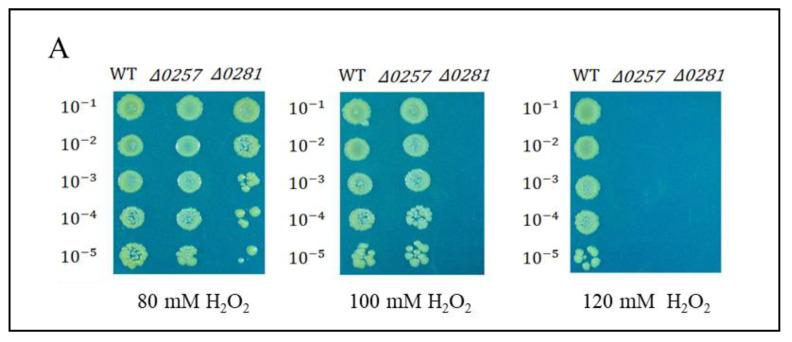
(**A**) Viability against H_2_O_2_ treatment at 80, 100, and 120 mM for 1 h was tested. Comparison of growth patterns among wild-type, Δ*dgeo*_0257, and Δ*dgeo*_0281 mutant strains. Growth was completely inhibited by 100 mM H_2_O_2_ at two different OD_600_ values of 0.5 (**B**) and 1.47 (**C**) in wild-type and Δ*dgeo*_0257 and Δ*dgeo*_0281 mutant strains. The OD values were determined by direct measurement.

**Figure 6 ijms-24-00469-f006:**
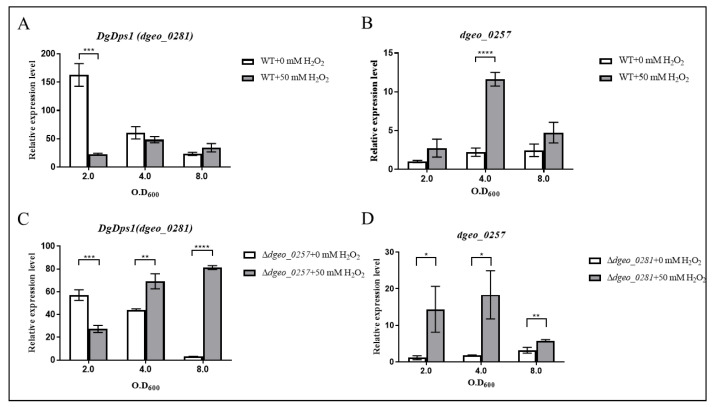
Expression levels of both DgDps1 (*dgeo*_0281) (**A**,**C**) and a putative Dps (*dgeo*_0257) (**B**,**D**) in the presence or absence of 50 mM H_2_O_2_ in the wild-type strain and in Δ*dgeo*_0257 and Δ*dgeo*_0281 mutants at three different growth phases, OD_600_ 2.0, 4.0, and 8.0, by qRT-PCR analysis, respectively. The probability *t*-test was considered significant at *p* < 0.05 (*), *p* < 0.01 (**), *p* < 0.001 (***), and *p* < 0.0001 (****).

**Figure 7 ijms-24-00469-f007:**
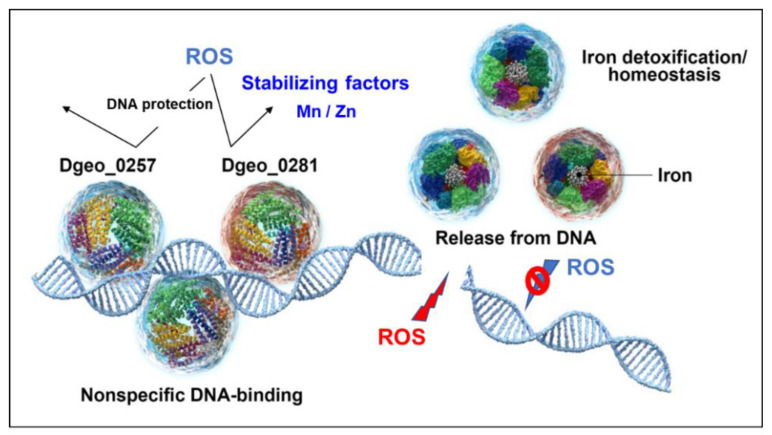
Regulatory scheme of DgDps1 (Dgeo_0281) and Dgeo_0257. Both DgDps1 and Dgeo_0257 proteins act as DNA protectants under oxidative stress conditions. DNA-binding stability was enhanced by Mn and Zn ions. When Dps proteins were released from DNA, iron ions were sequestrated to repress the Fenton reaction. However, free DNA damaged by ROS results in DNA cleavage.

## Data Availability

Not applicable.

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
