# Peer review of "Complementary Roles of Two DNA Protection Proteins from *Deinococcus geothermalis"

_ijms, 2022, doi:10.3390/ijms24010469_

Round 1

Reviewer 1 Report

The authors characterize two Dps proteins from Deinococcus geothermalis. From this work, authors deduce that one of them is an orthologous of the Deinococcus radidurans Dps protein, and the other, is a novel Dps protein that lacks the ferroxidase activity. The expression of the genes of these proteins under oxidative stress is complementary along the growth curve. Members of the Deinococcus genus survive oxidative stress and the role of the Dps protein in protecting DNA from this damage is crucial. The characterization of the two Dps proteins of D. geothermalis is interesting and deserves to be published. However, several aspects have to be taken into account.

General considerations:

-English should be thoroughly reviewed; some parts of the manuscript contain errors that make the text difficult to understand. Furthermore, some oversights in scientific language need to be corrected.

-The names of the two D. geothermalis Dps proteins characterized are confusing as the authors name the two proteins in two different ways that they use interchangeably throughout the entire manuscript. What's more, sometimes they use both names at the same time. If they want to use an abbreviation (DgDps1 and DgDps3 in place of Dgeo_0281 and Dgeo_0257, respectively) they should define it in the abstract and continue with it throughout the text, including figures and figure captions.

Specific considerations in order of appearance in the manuscript:

-Title only reflects the characterization of the Dgeo_0257 protein, which is important given the novelty of this protein, but from what can be deduced from the manuscript, the Dgeo_0281 protein have also been characterized for the first time in this work. A more informative title should be taken into consideration.

-In the first section of results, the figure S1 should be included in figure 1 with the rest of alignments. The three comparisons together help to understand main differences in secondary structure. In this section, the sentence from 121 to 123 is confusing, as it is not clear which protein lacks 40 aminoacids at the N-terminus. Please, rewrite this sentence and add the amino acid numbers guide in the alignments for more clarity.

-In the second results section, the sentence in line 148 "Both proteins were well induced..." is ambiguous and the reader cannot verify this information in the figure. Therefore, it can be removed.  Furthermore, in Figure S2, the meaning of "C" in the last lane of the Dgeo_0281 gel should be explained in the figure legend.

-Interpretation of the results from gel filtration based on Figure S3 is confusing. This figure has not an appropriate quality. Authors explain that the size of Dgeo_0257 in the absence of iron correspond to an icositethraedron (lanes 157 to 159), but in the last sentence (lanes 162 to 164) they write that the protein conformation changes in the presence of iron from the dodecameric form in the absence of iron. Given the lack of resolution to determine the size of proteins by gel filtration technique, a complementary technique as sucrose gradient sedimentation should be included in this work to confirm the size of the Dgeo_0257.

-Regarding EMSA assays, an explanation as to why the authors use the promoter and ORF sequence of dgeo_2112 should be included in the text. More information on the quantity and exact sizes of DNAs used for EMSA assays should appear at least in the materials and methods section. Information such as “approximately 250 nucleotides in length…”; “specific amounts…” (Rows 504 and 505); or “a long-distance promoter (Rows 168-169) is not admissible.

The 2.3 section of results is written in a confusing way (for example lines 171 to 173) and it is difficult to understand. It must be rewritten explaining in a detailed way all the results.

The sentence “The DNA binding behavior of Dgeo_0257 and Dgeo_0281 was different between the promoter region and the ORF region” (Lines 176 to 178) should be further explained. I do not see differences in the case of Dgeo_0257 in the promoter region and in the absence of iron and hardly any difference in the same situation for Dgeo_0281.

Authors do not say anything about the change of shift in the case of Dgeo_0257 bound to the promoter region in the presence of iron.

In the case of DNA protection assays, both proteins protect from oxidative stress induced by ferrous ions, but from Figure 3 it cannot be deduced that these proteins protect DNA from H2O2 treatment as the authors claim (Lines 197 to 198). The treatment with H2O2 in the presence of ferrous ions induce complete degradation of DNA in the presence of the proteins (Lanes 6 and 10). The figure does not show the control of this situation without protein, nor the H2O2 treatment without iron.

The section 2.4 of results is very difficult to understand. EMSAS are performed using ORF or Promoter regions interchangeably as template DNA with the different metals (except for Cs and Cr) and authors compare the results of both. I do not consider it is acceptable, more even when the authors considered different the binding behavior of both proteins in these two DNA templates (Lines 176 to 178). EMSAS must be performed under the same experimental conditions (the same DNA template) to compare the results. The EMSA with Dgeo_0281 in the presence of Cs with “fluctuating patterns” seems to be the result of a mistake. It does not seem to me an acceptable figure to be published.

Figure S4AB should be included into Figure 4 as the authors consider these results relevant.

-Regarding the physiological properties of mutants in section 2.5, I cannot see the 1-hour delay that author allude. Growth rates of mutants and wild-type strains should be calculated and indicated.

Figure S5 include the same results shown in Figure 5 C in a different panel way. It is redundant.

The names of the strains must be maintained in the different sections of Figure 5: Δdgeo_0257C and Δdgeo_0281C are the same that Δdgeo_0257/pRadgr_dgeo0257 and Δdgeo_0281/pRadgr_dgeo0281?.

Figure S6 does not appear mentioned in the manuscript

-In section 2.6, at line 272, to the sentence "However, DgDps1 expression was enhanced by OD600 4.0 (Fig. 6A)" should be added for better understanding "in the presence of H2O2".

To the sentence “Thus, DgDps1 took over the role of DgDps3 during the growth phase….” (Line 285) must also be added “under oxidative stress”.

-The Discussion section includes several information that must be in the Introduction section: Lines 303 to 307; 315 to 343; and 356 to 370.

-Paragraph 7 (Lines 371 to 378) corresponded to other work that has already been published.

-The Kd is the concentration at which 50% of the DNA is in a complex with the protein. This value for Dgeo_0257 cannot be deduced from Figure 2.

-Tables S1 and S2 correspond to a previous work (Lee C et al. 2019. Microrganisms). Please, remove them from this manuscript.

-Table S3 contains information of oligonucleotides used in the construction of strains Δdgeo_0257 and Δdgeo_0281, which has already been published (Lee C et al. 2019. Microrganisms). This information should be removed.

Author Response

General considerations:

-English should be thoroughly reviewed; some parts of the manuscript contain errors that make the text difficult to understand. Furthermore, some oversights in scientific language need to be corrected.

> The MS revision improved English correction by native speakers in the editing system.

-The names of the two D. geothermalis Dps proteins characterized are confusing as the authors name the two proteins in two different ways that they use interchangeably throughout the entire manuscript. What's more, sometimes they use both names at the same time. If they want to use an abbreviation (DgDps1 and DgDps3 in place of Dgeo_0281 and Dgeo_0257, respectively) they should define it in the abstract and continue with it throughout the text, including figures and figure captions.

 > Okay, accepting the reviewer's suggested opinion, the genetic name was retained in this MS because the Dps definition of the new estimated Dps protein dgeo_0257 is not enough to be identified in the current data (Reviewer 2 opinion). Thus, revised MS used Dgeo_0257 and DgDps1 for Dgeo_0281.

Specific considerations in order of appearance in the manuscript:

-Title only reflects the characterization of the Dgeo_0257 protein, which is important given the novelty of this protein, but from what can be deduced from the manuscript, the Dgeo_0281 protein have also been characterized for the first time in this work. A more informative title should be taken into consideration.

> I’m agree with reviewer opinion. The title was changed to “Complementary Roles of Two DNA Protection Proteins from Deinococcus geothermalis” for the DNA-binding and metal ion-sensing properties both DgDps1 and Dgeo_0257 proteins.

-In the first section of results, the figure S1 should be included in figure 1 with the rest of alignments. The three comparisons together help to understand main differences in secondary structure. In this section, the sentence from 121 to 123 is confusing, as it is not clear which protein lacks 40 amino acids at the N-terminus. Please, rewrite this sentence and add the amino acid numbers guide in the alignments for more clarity.

> Thank you for your comments. FIG. S1 moved to FIG. 1A to better explain amino acid sequence similarity and retention residues. The sentence in lines 121-123 has been changed.

-In the second results section, the sentence in line 148 "Both proteins were well induced..." is ambiguous and the reader cannot verify this information in the figure. Therefore, it can be removed.  Furthermore, in Figure S2, the meaning of "C" in the last lane of the Dgeo_0281 gel should be explained in the figure legend.

> We could induce both proteins using 0.5 ~ 1.5 mM isopropyl β-D-1-thiogalactopyranoside (IPTG). This data included in Fig S1A.

-Interpretation of the results from gel filtration based on Figure S3 is confusing. This figure has not an appropriate quality. Authors explain that the size of Dgeo_0257 in the absence of iron correspond to an icositethraedron (lanes 157 to 159), but in the last sentence (lanes 162 to 164) they write that the protein conformation changes in the presence of iron from the dodecameric form in the absence of iron. Given the lack of resolution to determine the size of proteins by gel filtration technique, a complementary technique as sucrose gradient sedimentation should be included in this work to confirm the size of the Dgeo_0257.

> Right, it not enough to determine the conformation of Dgeo_0257 in present gel-filtration analysis. Maybe for the multimeric conformation change, gel filtration using Superdex 200 column, the sucrose-gradient and size-diffraction analysis need to confirm. It is the further study contents for characteristics if novel Dps protein. Thus, we changed the sentence.

-Regarding EMSA assays, an explanation as to why the authors use the promoter and ORF sequence of dgeo_2112 should be included in the text. More information on the quantity and exact sizes of DNAs used for EMSA assays should appear at least in the materials and methods section. Information such as “approximately 250 nucleotides in length…”; “specific amounts…” (Rows 504 and 505); or “a long-distance promoter (Rows 168-169) is not admissible.

> 0257 is 10 µm and 0281 is about 3-folds more 30~45 µm in text. The DNA length for EMSA is added 250 nt in text.

The 2.3 section of results is written in a confusing way (for example lines 171 to 173) and it is difficult to understand. It must be rewritten explaining in a detailed way all the results.

> The DNA-protein complex of DgDps1 was located lower in the EMSA gel than Dgeo_0257-DNA complexes due to its larger size. EMSA data changed correctly and the explanation also revised.

The sentence “The DNA binding behavior of Dgeo_0257 and Dgeo_0281 was different between the promoter region and the ORF region” (Lines 176 to 178) should be further explained. I do not see differences in the case of Dgeo_0257 in the promoter region and in the absence of iron and hardly any difference in the same situation for Dgeo_0281.

> Thanks for your comments. The sentence was changed and EMSA data was added in Fig. 2. DNA-binding behaviors of DgDps1 and Dgo_0257 showed a similar pattern between the promoter region and the ORF region of randomly selected target gene, but the EMSA pattern was different due to DNA-binding and metal ions-sensing from two DNA regions.

Authors do not say anything about the change of shift in the case of Dgeo_0257 bound to the promoter region in the presence of iron.

  • The sentence is included Dgeo_0257 DNA-binding behavior.

In the case of DNA protection assays, both proteins protect from oxidative stress induced by ferrous ions, but from Figure 3 it cannot be deduced that these proteins protect DNA from H2O2 treatment as the authors claim (Lines 197 to 198). The treatment with H2O2 in the presence of ferrous ions induce complete degradation of DNA in the presence of the proteins (Lanes 6 and 10). The figure does not show the control of this situation without protein, nor the H2O2 treatment without iron.

>Right, H2O2 alone has not gross effect for DNA degradation even ferrous ion or ferrous ion plus hydrogen peroxide exhibits drastic DNA disappearance. DNA disappear depend on Fe(II) concentration but not affected by Fe(III) treatment. You can see below figure for DNA protection by the Dgeo_0257 when ferrous ion or ferrous ion plus H2O2 present.

The section 2.4 of results is very difficult to understand. EMSAS are performed using ORF or Promoter regions interchangeably as template DNA with the different metals (except for Cs and Cr) and authors compare the results of both. I do not consider it is acceptable, more even when the authors considered different the binding behavior of both proteins in these two DNA templates (Lines 176 to 178). EMSAS must be performed under the same experimental conditions (the same DNA template) to compare the results. The EMSA with Dgeo_0281 in the presence of Cs with “fluctuating patterns” seems to be the result of a mistake. It does not seem to me an acceptable figure to be published.

>There is shown the “fluctuation patterns” of EMSA and in vitro transcription assay in a sugar-sensing archaeal regulator through sugar concentration (Lee et al. 2007 Mol.Microbiol. 65:305-318). It is maybe sense molarity ratio between proteins and ions despite still it is unclear. EMSA assay was duplicated.

Figure S4AB should be included into Figure 4 as the authors consider these results relevant.

> Fig. S4AB included into Fig. 2CD in the revised MS and added explanation in text. It is much better for understanding. Thank you for your suggestion.

-Regarding the physiological properties of mutants in section 2.5, I cannot see the 1-hour delay that author allude. Growth rates of mutants and wild-type strains should be calculated and indicated.

>Right, the growth delay is less than 1 hour. The sentence was changed. In overall, Dgeo_0257 and DgDps1 mutant strains were revealed clear sensitivity against H2O2 by direct treatment in former supporting data. Reviewer 2 also recommended data exchange. Thus, Fig. 5 reorganized with viability and growth curves. The complementary data moved to supporting data.

Figure S5 include the same results shown in Figure 5 C in a different panel way. It is redundant.

>The experiments were separated in both. The supplementary data indicated direct measurement and treatment of H2O2 at OD 4.0 result in more dramatic different viability than saline washing and normalizing conditions for complementary strains. (check upper response)

The names of the strains must be maintained in the different sections of Figure 5: Δdgeo_0257C and Δdgeo_0281C are the same that Δdgeo_0257/pRadgr_dgeo0257 and Δdgeo_0281/pRadgr_dgeo0281?

>Right, the strain names were fixed as Δdgeo_0257/pRADgro_dgeo_0257 and Δdgeo_0281/pRADgro_dgeo_0281 for complementary strain in text of M&M section.

Figure S6 does not appear mentioned in the manuscript.

>Right, the revised MS not included Fig. S6 because the complementary strains have chloramphenicol resistance for selection of plasmid transformation. The growth was too delayed by the antibiotic. Nevertheless, complementary effects were revealed in physiological observation.

-In section 2.6, at line 272, to the sentence "However, DgDps1 expression was enhanced by OD600 4.0 (Fig. 6A)" should be added for better understanding "in the presence of H2O2".

>Thank you for your comment. The sentence was revised.

To the sentence “Thus, DgDps1 took over the role of DgDps3 during the growth phase….” (Line 285) must also be added “under oxidative stress”.

 >Thank you for your comment. The sentence was changed.

-The Discussion section includes several information that must be in the Introduction section: Lines 303 to 307; 315 to 343; and 356 to 370.

>Thank you for your comment. The several sentences were moved to introduction section.

-Paragraph 7 (Lines 371 to 378) corresponded to other work that has already been published.

>Right. The sentence was revised.

-The Kd is the concentration at which 50% of the DNA is in a complex with the protein. This value for Dgeo_0257 cannot be deduced from Figure 2.

>Right, the present data not correct calculated Kd value. Thus, the sentence was changed.

-Tables S1 and S2 correspond to a previous work (Lee C et al. 2019. Microrganisms). Please, remove them from this manuscript.

>Right, the table S1 and S2 were removed and the contents were mentioned in text for functional roles of Dps as a gene controller.

-Table S3 contains information of oligonucleotides used in the construction of strains Δdgeo_0257 and Δdgeo_0281, which has already been published (Lee C et al. 2019. Microrganisms). This information should be removed.

>Right, the construction of strain Δdgeo_0257 was deleted. Table S3 was modified with primer sets for EMSA in this revised MS Table S1.

Reviewer 2 Report

General comments

In the abstract section I am missing the manuscript aim. I suggest to direct the reader to the following parts stating something like that: “Ferritins are a large family of iron storage proteins. DNA-binding proteins from starved bacterial cells (Dps) are members of the ferritin family of proteins” Otherwise you are writing for aficionados about the protein binding in the presence or absence of ferrous and ferric ions. I am also missing the state of the art, I suggest including something like that: “Dgeo_0281 is annotated as DgDps1 DNA binding protein from starved cells and Dgeo_0257 is predicted to be a Dps-like protein . . .”

Specific Comments

1. Line (L) 10. Is Dgeo_0257 “novel” and “an orthologous of Dgeo_0281”? Two unrelated proteins may share up to 12% sequence identity. Dgeo_0257 shares only 11% sequence identity with Dgeo_0281 at the aa level. There is a little sequence identity even at their ferroxidase sites. How can the authors infer that both proteins have evolved from a common ancestor? Convergent evolution may apply. I can infer that Dgeo_0257 lacks a ferroxidase sites. The in vitro analyses of Dgeo_0281 do not support the hypothesis that His-tagged protein variant is a genuine Dps-like protein. I am afraid that His-Dgeo_0281 is a mutant variant.

2. L 16. The author stated “both purified proteins bound non-specifically to DNA”, the opposite is shown in Figure 2.

3. L 17 -21. I cannot reach the same conclusion that the authors. It is statedIn the presence of ferrous and ferric ions neither Dgeo_0257 nor Dgeo_0281 could readily bind to DNA” contradicting the data presented in Figure 2. And “Mutants in which the Dps gene was disrupted exhibited higher sensitivity to oxidative stress than the wild-type strainHow much is “higher”. I see no significant differences, please quantify and rephrase. It is stated “both proteins exhibited more stable DNA-binding in the presence of zinc and manganese ions”. I cannot agree with the presented information, because there is a 3-fold difference in their respective Kapp between Fig. 2 and Fig. 3 (lane 3) and the experimental difference in the presence of Mn+2 or Zn+2 is within the experimental error between the different figures.

4. L 21-22. Please rephrase overstatements such as “expression levels of each gene showed an opposite correlation with each other under H2O2 treatment conditions

5. L 106 - 108. I cannot see the supplementary Figures. It is not stated why did the authors assume that Dgeo_0257 is a Dps-like protein. The structural prediction should be documented and compared with Dgeo_0281 or with any genuine Dps protein.

6. L 122 -123. DgDps1 only shows 11% sequence identity (a value considered as random identity). Please highlight the ferroxidase center on DgDps1. Dgeo_0257 lacks the conserved ferroxidase center. Why did the authors consider that Dgeo_0257 is a DNA-protecting Dps-like protein? The expression of Dgeo_0257 is unlinked with nutritional deprivation, a signature of Dps-like protein are missing.

7. Figure 1. The meaning of the red boxes is not explained. The DNA-binding domain of DgDps1 should be indicated. Please use the AlphaFold program to predict the structure of Dgeo_0257 and compare it with the one of DgDps1 (Dgeo_0281). A sequence identity above 25% is necessary to predict a similar protein fold.

8. L 123-124. Please rephrase the sentence. The text in the Figure legend is clearer that in the main text.

9. L 144 – 145. Did authors check whether the His-tagged Dgeo_0281 or Dgeo_0257 clones can complement their null mutants?

10. L 157 – 162. I am confused. How can the authors deduce an icositetrahedron structure by gel filtration even using a Superdex 75 column? From our experience Superdex 75 is a column with a separation range for molecules with molecular masses between 3.000 and 70.000 Da, above 100.000 Da there is no linearity. How is it deduced that “Dgeo_0257 157 was approximately 480 kDa, which corresponded to an icositetrahedron form”? What is the predicted molecular mass of a protein that “moved to the posterior region”. Can the authors discriminate a 480 kDa from 240 kDa with a Superdex 75 column? The 440 kDa marker may peak at fraction 49, the 75 kDa at fraction 54, and the 43 kDa at fraction 68. In the absence of Fe ions Dgeo_0257 peaks at fraction 51 (<440 kDa), in the presence of 1 mM Fe (II) peaks at fraction 56 (<75 kDa) and in the presence of Fe (III) the protein fraction is significant reduced and peaked at fraction 50 (<440 kDa) and 54 (>75 kDa), vwith a low molecular mass peak. How can the authors conclude that Dgeo_0257 is a 24-mer (expected 550 kDa) with icositetrahedron structure in the absence of Fe ions? What are the deduced molecular masse of Dgeo_0257 in presence of Fe (II) and Fe (III)?

The 75 kDa at fraction 54, the 43 kDa at fraction 68 and the 29 kDa at fraction 78. In the absence of Fe ions DgDps1 peaks at fractions 64 (>43 kDa) and 71 (>29 kDa), in the presence or absence of Fe ions. How can the authors conclude that DgDps1 is in a momomer (expected 23 kDa) -dimer (expected 46 kDa) equilibrium?

11. Figure 2 and L 166-170. It is generally believed that Dps proteins may exhibit some sequence or structural selectivity, but the authors assumed that they have non-specific DNA-binding activity. The data presented in Fig. 2 support the first assumption. What is the DNA concentration used? What “super-shift” means in this context. The apparent binding constant (Kapp) of Dgeo_0257 (24-mer) is about 4 and 3 uM and DgDps1 (mono-dimer) about 45 and 60 uM for ORF and promoter DNA, respectively, in the absence of Fe+2. Conversely, in the presence of Fe+2, the Dgeo_0257 (12-mer) Kapp is >10 and about 5 uM and DgDps1 (mono-dimer) Kapp is 22 and >60 uM for ORF and promoter DNA, respectively. Since the Dps proteins bind to and condense DNA, but DgDps1 forms a discrete product of low molecular mass complex in the presence or absence of Fe+2, I wonder whether the His-tag is not altering the DgDps1 activity. Furthermore, the mobility of mono-dimer DgDps1- ORF DNA or dodecamer Dgeo_0257-promoter DNA forms a discrete of similar molecular mass in the presence Fe+2, I wonder about the protein-DNA stoichiometry. What is the difference between the ORF and promoter DNA substrate?

12. Figure 3 and L 192 – 193. In Fig. 2A the Kapp of Dgeo_0257 and DgDps1 for ORF DNA is >10 uM and <30 uM, respectively, in the presence of Fe+2. In Fig. 3 and presence of Fe+2, the protein concentrations used are below and above Kapp for Dgeo_0257 and DgDps1, respectively. Does Dgeo_0257, below kapp, protects the ORF DNAs from Fe+2 oxidative stress? Or Dgeo_0257 binds and oxidizes iron to prevent the formation of oxidative free radicals. Similar model applies for DgDps1. Under the experimental conditions used environmental O2 oxidase Fe (II). Does Dgeo_0257 or DgDps1 H2O2 to oxidize Fe(II) to Fe (III)? DrDps1 has distinct iron exit channels that constantly release Fe(II) and contribute to DNA damage. Why H2O2 was not included in the assay.

13. L 206 – 214. Please include Supplementary Figure S4 as part (B) in Figure 3is not in the Supplementary file

14. L 214- 216. The authors stated “Interestingly, as the concentrations of manganese and zinc ions increased, the two Dps proteins were much better bound to DNA, which further stabilized DNA binding (Fig. 4).” The sentence contradicted the results presented in Figure 2. Here, the Dgeo_0257 Kapp for ORF DNA is about 3 uM, and the reaction is saturated at 5 uM. The data of Fig. 2 are not reproduced in Fig. 3 (line 3). Here, the Kapp is about 3-fold lower than in Fig. 2. The differences are below the data dispersion between the figures.

15. L 239- 243. Cannot be evaluated. The Figure is missing.

16. L 243-253. Why are Figure 5B and supplementary Fig. S6 so different? Please replace Figure 5C because I cannot see any significative differences. By the supplementary Figure S5C and modify the text. Addition of chloramphenicol increased cell survival only in the Δdgeo_0281, but not in Δdgeo_0257. How do the authors interpret these results?

17. L 267 – 270. Dps is defined as an abundant protein in starved cells (e.g., stationary phase cells). Why the authors are taken The expression level of dgeo_0257 at OD600 2.0 [early exponential phase as depicted in Fig. 5B] in the wild-type strain was taken as the control.” This is distorting the assay. Please show the crude data. Is the expression level of DgDps1 “upregulated”? I can deduce that, at low cell density, promoter utilization of the dps1 gene is 150-fold higher than that of dgeo_0257. In the wt strain, the expression level of dgeo_0257 did not significantly vary at low and high cell density. Finally, dgeo_0257 is poorly expressed at stationary phase

18. The Discussion section is to evaluate the obtained results and if necessary, to compare them with similar or different published results, rather than to make a second introduction.

19. L 382-383. The statement “However, a novel DgDps3 exhibited higher DNA-binding affinity and 382 more metal ion sensitivity than DgDps1” remains to be experimentally confirmed, by showing that the His-tag does not compromise the protein activity.

20. L 387-390. How can the authors assume thatIn the presence of H2O2, the DgDps1 levels of the DgDps3-disrupted mutant were gradually induced with cell growthif the expression level of dgeo_0257 at OD600 2.0 in the wild-type strain was taken as the control.

21. L 402-404. The statement “The novel Dps protein DgDps3 (Dgeo_0257) has higher DNA-binding activity than DgDps1 (Dgeo_0281) and sense to ferric and ferrous ions result in DNA releasing with similar behavior to DgDps1cannot be deduced from the presented results. Gel filtration reveals, that DgDps1 is in a monomer-dimer equilibrium in the presence of absence of Fe ions, but the oligomeric state of Dgeo_0257 is altered. The authors must show that His-DgDps1 can fully complement a Δdgeo_0281 mutant strain.

22. L 404-405. In the presence of Mn2+ or Zn2+, there are saturating concentrations of Dgeo_0257 and nothing can be deduced from those results. Conversely, DgDps1 was not apparently affected, I guess that the poor resolution of the gel distorts the interpretation, please underexpose the gel, and I guess that both free and protein-DNA complexes can be seen.

Miscellaneous

1. L 73. Please delete one of the words underlines. “However, although the basic mechanisms of iron entry . . .”.

2. L 121-123. Please rephrased indicating the sequence identity of the N-terminal region.

2. L 152. I am not able to understand the sentence “90% purity per liter of culture”. Are the authors indicating that due to high overexpression 90% of total proteins, in the cell batch, are Dgeo_0281 or Dgeo_0257? Or are they stating that the proteins are 90% pure?

3. 168. What is the meaning ofa long-distance promoter and ORF region.” For a protein that binds DNA in a sequence independent manner, but with certain structural selectivity.

4. L 239. Please rephrase “both Dps genes are not essential”. It has not been shown that dgeo_0257 is a Dps-like protein.

Author Response

Reviewer 2.

Comments and Suggestions for Authors

General comments

In the abstract section I am missing the manuscript aim. I suggest to direct the reader to the following parts stating something like that: “Ferritins are a large family of iron storage proteinsDNA-binding proteins from starved bacterial cells (Dps) are members of the ferritin family of proteins” Otherwise you are writing for aficionados about the protein binding in the presence or absence of ferrous and ferric ions. I am also missing the state of the art, I suggest including something like that: “Dgeo_0281 is annotated as DgDps1 DNA binding protein from starved cells and Dgeo_0257 is predicted to be a Dps-like protein . . .”

 >Thank for your comments. Right, Dps is a bacterioferritin-like protein which was incorporated with iron ions for iron homeostasis and detoxification. Thus, based on this knowledge, we performed EMSA and gel filtration effects of iron. The abstract and many sentences were revised following your comments and suggestions. Actually, the present data not enough confidence for protein conformation for example, a typical Dps revealed 24-mer form or 12-mer form from other bacteria for example, E. coli and D. radiodurans as well studied a model organism in genus Deinococcus. However, interestingly, an orthologous Dps protein Dgeo_0281 (contrast to Dps1 of D. radiodurans) revealed monomer or dimer form from superdex 75 chromatograpy. You are right! The separation range of Superdix 75 is limited to 70 kDa. Thus, the conformation of Dgeo_0257 can’t determine in present data. Thus, we changed the sentence to multimeric form shift (See below author responses). DgDps1 was not induced growth phase-dependent manner in H2O2 absent condition, generally Dps is a dominant protein in stationary phase as like the name origin. In these results, we characterized D. geothermalis Dps ortholog protein comparing a novel putative Dps protein Dgeo_0257. The genome annotation information indicated that Dgeo_0257 is a putative Dps. However, the protein has not a conserved ferroxidase active site and the amino acid sequence identity is lower than 20%. At the moment we do not decided coming from convergent evolution or same ancestor origin. Nevertheless, our questions are whether the possible Dps function of Dgeo_0257 in genus Deinococcus. We purified both Dps candidate proteins, analyzed protein naïve conformation, DNA-binding specificity, and various metal ions sensitivity. There is an important finding came from qRT-PCR analysis. That is both proteins are complementary functions in each gene deficient condition under the oxidative stress conditions. Thus, the MS included significant results for promoting this research area further studies detail protein conformation, DNA-binding affinity, metal ion-sensing and interplay between DgDps1 and Dgeo_0257.

Specific Comments

  1. Line (L) 10. Is Dgeo_0257 “novel” and “an orthologous of Dgeo_0281”? Two unrelated proteins may share up to 12% sequence identity. Dgeo_0257shares only 11% sequence identity with Dgeo_0281 at the aa level. There is a little sequence identity even at their ferroxidase sites. How can the authors infer that both proteins have evolved from a common ancestor? Convergent evolution may apply. I can infer that Dgeo_0257 lacks a ferroxidase sites. The in vitro analyses of Dgeo_0281 do not support the hypothesis that His-tagged protein variant is a genuine Dps-like protein. I am afraid that His-Dgeo_0281 is a mutant variant.

 > The revised MS indicated that the Dgeo_0257 is a putative Dps protein and DgDps1 (Dgeo_0281) is an orthologous DrDps1 protein. Still we do not mention that both proteins are paralogous relationship from convergent evolution or same ancestor origin because both protein amino acid sequence similarity is relatively low.

Right, the his-tagged extension maybe be affected the typical conformation for a shell-like protein assembly. It is really important point for further study.

  1. L 16. The author stated “both purified proteins bound non-specifically to DNA”, the opposite is shown in Figure 2.

 > In general, DNA-binding specificity was clearly revealed on EMSA in many cases. However, here both proteins revealed similar DNA-binding affinity on two target DNA regions such as promoter and Orf regions each other. Right, DNA-binding behavior somehow different on iron present conditions. For example, free DNA positions are different on present increasing concentration iron between Fig. 2B and 2C. Nevertheless, both proteins revealed very similar DNA-binding in two different target DNAs.

  1. L 17 -21. I cannot reach the same conclusion that the authors. It is stated “In the presence of ferrous and ferric ions neither Dgeo_0257 nor Dgeo_0281 could readily bind to DNA” contradicting the data presented in Figure 2. And “Mutants in which the Dps gene was disrupted exhibited higher sensitivity to oxidative stress than the wild-type strain” How much is “higher”. I see no significant differences, please quantify and rephrase. It is stated “both proteins exhibited more stable DNA-binding in the presence of zinc and manganese ions”. I cannot agree with the presented information, because there is a 3-fold difference in their respective Kapp between Fig. 2 and Fig. 3 (lane 3) and the experimental difference in the presence of Mn+2 or Zn+2 is within the experimental error between the different figures.

  > The several sentences, you pointed out in abstract were not changed because the additional data, for example Fig. 2C, indicated that the increased concentration of ferrous ions affected to DNA-protein complexes result in DNA-protein dissociation. Fig. 5 was also changed from supporting data following your suggestion. It was supported the difference of sensitivity between WT and mutants. Thank you.

  1. L 21-22. Please rephrase overstatements such as “expression levels of each gene showed an opposite correlation with each other under H2O2 treatment conditions

 > The sentence revised to “Furthermore, the expression levels of each gene showed an opposite correlation under H2O2 treatment conditions.”.

  1. L 106 - 108. I cannot see the supplementary Figures. It is not stated why did the authors assume that Dgeo_0257 is a Dps-like protein. The structural prediction should be documented and compared with Dgeo_0281 or with any genuine Dps protein.

> Fig. S1 moved to Fig. 1A for comparison between DgDps1 and DrDps1. Actually, the genome annotation information indicated that Dgeo_0257 is a putative Dps. However, the protein has not a conserved ferroxidase active site and amino acid sequence identity is lower than 20%. Thus, we characterized Dps properties for DNA-binding and metal ion-sensing especially ferrous ion in this work.

  1. L 122 -123. DgDps1 only shows 11% sequence identity (a value considered as random identity). Please highlight the ferroxidase center on DgDps1. Dgeo_0257 lacks the conserved ferroxidase center. Why did the authors consider that Dgeo_0257 is a DNA-protecting Dps-like protein? The expression of Dgeo_0257 is unlinked with nutritional deprivation, a signature of Dps-like protein are missing.

 > Amino acid sequence alignment between DgDps1 and DrDps1 (revised Fig. 1A) or DgDps1 and DgDps3 (revised Fig. 1C) marked secondary structure and ferroxidase active center. General Dps was induced at stationary phase. Thus, we testified three growth points especially OD 8.0 is entered to stationary phase in D. geothermalis. D. radiodurans also has an orthologous protein of Dgeo_0257 that is Dr0582 with 70% amino acid identity as a hypothetic protein.

  1. Figure 1. The meaning of the red boxes is not explained. The DNA-binding domain of DgDps1 should be indicated. Please use the AlphaFold program to predict the structure of Dgeo_0257 and compare it with the one of DgDps1 (Dgeo_0281). A sequence identity above 25% is necessary to predict a similar protein fold.

 > In revised Fig. 1B, red-marked region indicated a possible DNA-binding domain from prediction program Prabi. From Fig. 1C red box indicated the conserved ferroxidase active sites in DgDps1. However, Dgeo_0257 has not shown these conserved amino acid residues for ferroxidase activity. We also performed protein structure prediction using by Alphafold program. There are four helixes bundle but N- and C-terminal extension and secondary structures are quite different between DgDps1 and DgDps3. At the moment we do not decided both proteins coming from convergent evolution or same ancestor origin.

  1. L 123-124. Please rephrase the sentence. The text in the Figure legend is clearer that in the main text.

 > Fig. 1A added from supporting data and the sentence revised.

  1. L 144 – 145. Did authors check whether the His-tagged Dgeo_0281 or Dgeo_0257 clones can complement their null mutants?

> For the complementary assay of target gene-deficient mutants, we used pRADgro vector for general gene expression system in genus Deinococcus. The His-tagged recombinant protein vector used for protein purification in heterologous E. coli. I guess it is general approach for the complementation and protein induction and purification.

  1. L 157 – 162. I am confused. How can the authors deduce an icositetrahedron structure by gel filtration even using a Superdex 75 column? From our experience Superdex 75 is a column with a separation range for molecules with molecular masses between 3.000 and 70.000 Da, above 100.000 Da there is no linearity. How is it deduced that “Dgeo_0257 157 was approximately 480 kDa, which corresponded to an icositetrahedron form”? What is the predicted molecular mass of a protein that “moved to the posterior region”. Can the authors discriminate a 480 kDa from 240 kDa with a Superdex 75 column? The 440 kDa marker may peak at fraction 49, the 75 kDa at fraction 54, and the 43 kDa at fraction 68. In the absence of Fe ions Dgeo_0257 peaks at fraction 51 (<440 kDa), in the presence of1 mM Fe (II) peaks at fraction 56 (<75 kDa) and in the presence of Fe (III) the protein fraction is significant reduced and peaked at fraction 50 (<440 kDa) and 54 (>75 kDa), vwith a low molecular mass peak. How can the authors conclude thatDgeo_0257 is a 24-mer (expected 550 kDa) with icositetrahedron structure in the absence of Fe ions? What are the deduced molecular masse of Dgeo_0257 in presence of Fe (II) and Fe (III)?

> You are right! The separation range of Superdix 75 is limited to 70 kDa. Thus, the conformation of Dgeo_0257 can’t determine in present data. Thus, we changed the sentence to multimeric form shift. The elution times of protein were written in text.

The 75 kDa at fraction 54, the 43 kDa at fraction 68 and the 29 kDa at fraction 78. In the absence of Fe ions DgDps1 peaks at fractions 64 (>43 kDa) and 71 (>29 kDa), in the presence or absence of Fe ions. How can the authors conclude that DgDps1 is in a momomer (expected 23 kDa) -dimer (expected 46 kDa) equilibrium?

>Fig. S2 was modified the correct positions of standard size markers and shown in inner box. DgDps1 exhibits about 35 kDa size in SDS-PAGE data (newly added Fig. S1A). Thus, the conformation of DgDps1 was relatively corrected.

  1. Figure 2 and L 166-170. It is generally believed that Dps proteins may exhibit some sequence or structural selectivity, but the authors assumed that they have non-specific DNA-binding activity. The data presented in Fig. 2 support the first assumption. What is the DNA concentration used? What “super-shift” means in this context. The apparent binding constant (Kapp) of Dgeo_0257 (24-mer) is about 4 and 3 uM and DgDps1 (mono-dimer) about 45 and 60 uM for ORF and promoter DNA, respectively, in the absence of Fe+2. Conversely, in the presence of Fe+2, the Dgeo_0257 (12-mer) Kapp is >10 and about 5 uM and DgDps1 (mono-dimer) Kapp is 22 and >60 uM for ORF and promoter DNA, respectively. Since the Dps proteins bind to and condense DNA, but DgDps1 forms a discrete product of low molecular mass complex in the presence or absence of Fe+2, I wonder whether the His-tag is not altering the DgDps1 activity. Furthermore, the mobility of mono-dimer DgDps1- ORF DNA or dodecamer Dgeo_0257-promoter DNA forms a discrete of similar molecular mass in the presence Fe+2, I wonder about the protein-DNA stoichiometry. What is the difference between the ORF and promoter DNA substrate?

 > Thank you for your detail comments for the apparent DNA-binding constant. Based on conformational information between Dgeo_0257 and DgDps1, EMSA data also exhibit DNA-protein complexes on gels.

  1. Figure 3 and L 192 – 193. In Fig. 2A the Kapp of Dgeo_0257 and DgDps1 for ORF DNA is >10 uM and <30 uM, respectively, in the presence of Fe+2. In Fig. 3 and presence of Fe+2, the protein concentrations used are below and above Kapp for Dgeo_0257 and DgDps1, respectively. Does Dgeo_0257, below kapp, protects the ORF DNAs from Fe+2 oxidative stress? Or Dgeo_0257 binds and oxidizes iron to prevent the formation of oxidative free radicals. Similar model applies for DgDps1. Under the experimental conditions used environmental O2 oxidase Fe (II). Does Dgeo_0257 or DgDps1 H2O2 to oxidize Fe(II) to Fe (III)? DrDps1 has distinct iron exit channels that constantly release Fe(II) and contribute to DNA damage. Why H2O2 was not included in the assay.

> Right, Kd and Kapp values were not constant in present EMSA data set. Thus, we explained mildly the Kd > 4 uM of Dgeo_0257 and Kd > 40 uM of DgDps1. For the DNA protection by both proteins, you can see below figure for DNA damage by H2O2 treatment. Actually, in H2O2 alone, DNA damage does not big affected.

  1. L 206 – 214. Please include Supplementary Figure S4 as part (B) in Figure 3is not in the Supplementary file

 > Thank you for your comments. Fig. 2CD was changed from supporting data. I guess it is enough to explanation for metal-sensing behavior.

  1. L 214- 216. The authors stated “Interestingly, as the concentrations of manganese and zinc ions increased, the two Dps proteins were much better bound to DNA, which further stabilized DNA binding (Fig. 4).”The sentence contradicted the results presented in Figure 2. Here, the Dgeo_0257 Kapp for ORF DNA is about 3 uM, and the reaction is saturated at 5 uM. The data of Fig. 2 are not reproduced in Fig. 3 (line 3). Here, the Kapp is about 3-fold lower than in Fig. 2. The differences are below the data dispersion between the figures.

 > In present data, we could not say exact Kd and Kapp values of DNA-binding of both proteins. You are right, each experimental condition was modified such as you pointed out. For the exact calculation of the kinetic study, we need further experiments. Please see the tendency of action mode between Dgeo_0257 and DgDps1.

  1. L 239- 243. Cannot be evaluated. The Figure is missing.

 > Fig. 5 was changed. The sentence revised.

  1. L 243-253. Why are Figure 5B and supplementary Fig. S6 so different? Please replace Figure 5C because I cannot see any significative differences. By the supplementary Figure S5C and modify the text. Addition of chloramphenicol increased cell survival only in the Δdgeo_0281, but not in Δdgeo_0257. How do the authors interpret these results?

 > Fig. S5 replaced to Fig. 5. Actually, the experiments are different approaches. In present Fig. 5 data were performed direct H2O2 treatment at the pointed growth phases. The complementary experiments performed in indirect approaches including dilute OD measurement after washing and OD normalizing. A little differently handled experiments but revealed quite different sounds.

  1. L 267 – 270. Dps is defined as an abundant protein in starved cells (e.g., stationary phase cells). Why the authors are taken “The expression level of dgeo_0257 at OD600 2.0 [early exponential phase as depicted in Fig. 5B] in the wild-type strain was taken as the control.” This is distorting the assay. Please show the crude data. Is the expression level of DgDps1 “upregulated”? I can deduce that, at low cell density, promoter utilization of the dps1 gene is 150-fold higher than that of dgeo_0257. In the wt strain, the expression level of dgeo_0257 did not significantly vary at low and high cell density. Finally, dgeo_0257 is poorly expressed at stationary phase

 > Generally, Dps proteins were induced at stationary phase and oxidative stress conditions. However, many cases of antioxidant responses were depending on intracellular redox-imbalances. Thus, the expression levels of DgDps1 and Dgeo_0257 genes were detected on three different growth points and performed. Interestingly, DgDps1 is a dominant protein at early exponential phase of OD=2.0. This related expression level came comparing with Dgeo_0257 level of 1.0 in WT OD=2.0. This finding need to further experiments for protein levels using western blots in growth phases or using different control such as a constitute expressed gene products.

  1. The Discussion section is to evaluate the obtained results and if necessary, to compare them with similar or different published results, rather than to make a second introduction.

 > The several sentences were moved to introduction part. For examples, general properties and Dps information in three cases of bacteria were introduced such as Vibrio, Nocostoc, and D. radiodurans.

  1. L 382-383. The statement “However, a novel DgDps3 exhibited higher DNA-binding affinity and 382 more metal ion sensitivity than DgDps1” remains to be experimentally confirmed, by showing that the His-tag does not compromise the protein activity.

 > Right, we do not know that His-tag effects on DNA-binding and metal-sensing. For the detail measurement of protein activity, we need to His-tag omitted version protein however, unfortunately here we compared both His-tagged protein for DNA-binding and metal-sensing on EMSA.

  1. L 387-390. How can the authors assume that “In the presence of H2O2, the DgDps1 levels of the DgDps3-disrupted mutant were gradually induced with cell growth” if the expression level of dgeo_0257 at OD600 2.0 in the wild-type strain was taken as the control.

 > Right, the expression level of DgDps1 much higher than Dgeo_0257 in WT. Thus, we controlled the normalizing with WT OD=2.0 of Dgeo_0257. Then, calculated DgDps1 expression level. When H2O2 present condition was used, in 50 mM H2O2 present condition, WT OD=2.0 of Dgeo_0257 level is the control for 1.0.

  1. L 402-404. The statement “The novel Dps protein DgDps3 (Dgeo_0257) has higher DNA-binding activity than DgDps1 (Dgeo_0281) and sense to ferric and ferrous ions result in DNA releasing with similar behavior to DgDps1” cannot be deduced from the presented results. Gel filtration reveals, that DgDps1 is in a monomer-dimer equilibrium in the presence of absence of Fe ions, butthe oligomeric state of Dgeo_0257 is altered. The authors must show that His-DgDps1 can fully complement a Δdgeo_0281 mutant strain.

 > The sentence revised to “The putative Dps protein Dgeo_0257 has over six-fold higher DNA-binding activity than that of DgDps1 and sense to ferric and ferrous ions result in DNA releasing with similar behavior to DgDps1 (Fig. 2).”. DgDps1 complement experiment exhibits stronger viability than WT in Fig. 5C but Dgeo_0257 complement does not enhance on viability. Thus, we need to further study for Dgeo_0257 physiological properties such as DNA-protection and antioxidant response.

  1. L 404-405. In the presence of Mn2+ or Zn2+, there are saturating concentrations of Dgeo_0257 and nothing can be deduced from those results. Conversely, DgDps1 was not apparently affected, I guess that the poor resolution of the gel distorts the interpretation, please underexpose the gel, and I guess that both free and protein-DNA complexes can be seen.

 > In Fig. 4, both DgDps1 and Dgeo_0257 proteins exhibit better DNA-binding when Mn and Zn present conditions. In the case of DgDps1 revealed novel DNA-protein complex with Mn ion but not Zn ion. Added arrows marking in Fig. 4.

Miscellaneous

  1. L 73. Please delete one of the words underlines. “However, although the basic mechanisms of iron entry. . .”.

  > The sentence revised to “However, although the basic mechanisms of iron entry and oxidation have been characterized, ….”.

  1. L 121-123. Please rephrased indicating the sequence identity of the N-terminal region.

  > The sentence revised to correct Fig. 1 and added “In present, we can’t decide whether both DgDps1 and Dgeo_0257 came as convergent evolutionary products and had a same ancestor.”.

N-terminal 40 aa-length extension of Dgeo_0257 was deleted because recent ORF prediction and the alphafold prediction also not illustrated.

  1. L 152. I am not able to understand the sentence “90% purity per liter of culture”. Are the authors indicating that due to high overexpression 90% of total proteins, in the cell batch, are Dgeo_0281 or Dgeo_0257? Or are they stating that the proteins are 90% pure?

  > The sentence revised to “approximately 90% purity after Ni-NTA affinity chromatography ….”.

  1. 168. What is the meaning of “a long-distance promoter and ORF region.” For a protein that binds DNA in a sequence independent manner, but with certain structural selectivity.

  > The sentence revised to “with a ca. 250 nt-length promoter and ORF region.”. Your right. Some DNA-binding proteins have not recognized a specific DNA sequences of inverted repeat somehow bound to non-specific sequence manner. It is maybe certain structural selectivity. For example, in my experience, archaeal regulator TrmBL2 bound to DNA with non-specific DNA sequence recognition but structural selectivity (Lee et al., 2007 MM : TrmBL2 2015 paper).

  1. L 239. Please rephrase “both Dps genes are not essential”. It has not been shown that dgeo_0257 is a Dps-like protein.

> I accepted your opinion that dgeo_0257 has not enough confidence for Dps. Dgeo_0257 is explained a putative Dps protein. The sentence revised to “both a DgDps1 and a putative Dps protein, Dgeo_0257 are not essential.”.

Round 2

Reviewer 1 Report

The revised manuscript with the suggested changes has been markedly improved. I only have a couple of minor considerations:

-Please, review figure 3, in the legend you indicated “4-5 & 8-9, increased Dgeo_0257 and DgDps1 proteins with ferric ions of 0.5 mM” (Line 246), but Ferrous ion appears in the figure.

-The last sentence of the Conclusion section is not clear to me: “Therefore, the expressional controls of both Dps are growth phase-dependent regulation and complementary to each other.” (lines 419-20). Perhaps you would like to say: “Therefore, the transcriptional regulation of both Dps is growth phase-dependent and complementary to each other”

Author Response

Thank you for your meticulous and accurate opinion. The final version has been revised according to the kind opinion of you. Your comment contributed greatly to the completion of the paper.